# Research on the Measurement and Influencing Factors of Carbon Emissions in the Swine Industry from the Perspective of the Industry Chain

**Yaguai Yu [1,2], Qiong Li [1,*], Yinzi Bao [1], Ersheng Fu [1], Yiting Chen [1] and Taohan Ni [3]**

1   Business School, Ningbo University, Ningbo 315211, China; yuyaguai@nbu.edu.cn (Y.Y.);
    baoyinzi00@sina.cn (Y.B.); 18868530403@163.com (E.F.); chenyiting202210@126.com (Y.C.)
2   Donghai Academy, Ningbo University, Ningbo 315211, China
3   Business School, University of Nottingham, Ningbo 315199, China; alexnee2004@126.com
*   Correspondence: liqiong971026@126.com; Tel.: +86-18858248460

**Abstract:** From the perspective of the industry chain, this paper uses the life cycle assessment (LCA) method to divide the swine industry into six production stages: feed crop cultivation, feed crop transportation and processing, intestinal fermentation, manure management, energy consumption in pig farming, and slaughtering and processing. Using the LCA method, the carbon emissions from the swine industry are measured from 2001 to 2020 for the whole country and 31 provincial regions. Based on the measurement results, this paper analyzes the dynamic evolution of carbon emissions from the national swine industry during the study period. Meanwhile, the spatial divergence in carbon emissions from the swine industry and the share of carbon emissions from each production stage were further analyzed by combining different provincial regions and production stages. Afterward, this paper uses the Logarithmic Mean Divisia Index (LMDI) model to decompose the influencing factors of carbon emissions at the national and provincial levels, and in each production stage. It is found that (1) The dynamic evolution of China's swine industry carbon emissions from 2001 to 2020 roughly follows a trend of "slow growth—sharp decline—fluctuating rise—fluctuating decline." The fluctuations are influenced by multiple factors, including the industry structure, agricultural policy, and farming scale. The primary driver for the increase in carbon emissions from the swine industry is the growth in demand for pork consumption, leading to the rise in swine supply. (2) In terms of spatial divergence at the provincial level, the regional differences in carbon emissions from the swine industry are significant, the total carbon emissions and unit carbon emissions of Jiangsu, Anhui, and Henan are higher than the national average. (3) In the production stages of the swine industry, feed crop cultivation and manure management are the primary sources of carbon emissions, associated with factors such as substantial feed consumption, crop production patterns, and backward manure management practices. (4) Regarding influencing factors, production efficiency, industry structure, and urbanization level have inhibiting effects on carbon emissions in the swine industry. Economic development and population scale have promoting effects. Production efficiency is the most significant inhibiting factor, and economic development is the most significant promoting factor. Finally, suggestions are made to curb carbon emissions in China's swine industry, including strengthening environmental control, formulating long-term plans for carbon emission reduction, delineating key areas and demonstration bases for carbon emission reduction, enhancing expertise in fertilizer application and manure treatment, and improving agricultural machinery and equipment.

**Keywords:** swine industry; carbon emissions; decomposition of influencing factors; life cycle assessment; LMDI decomposition model; industry chain

## 1. Problem Formulation and Literature Review

### 1.1. Problem Formulation

With escalating global warming, countries worldwide have reached a consensus on reducing carbon emissions and developing a green economy. To collectively address climate change, China has solemnly committed to the international community, aiming to peak carbon emissions before 2030 and achieve carbon neutrality by 2060 [1]. Introducing the "Dual Carbon" goals presents both opportunities and challenges for China.

While the primary sources of global carbon emissions are the secondary and tertiary industries, carbon emissions from agricultural production activities are also significant. Statistics indicate that from 2007 to 2016, carbon emissions from the agricultural food system accounted for 21% to 37% of the total global carbon emissions, with livestock contributing a substantial 18% [2]. Additionally, a report from the Food and Agriculture Organization of the United Nations (FAO) during the COP26 climate summit in November 2021 revealed that in 2019, human-induced $CO_2$ emissions were 54 billion tonnes, with 17 billion tonnes of $CO_2$ equivalent originating from agriculture and food systems [3]. Furthermore, the FAO emphasized in the book "Livestock's Long Shadow: Environmental Issues and Options" that the greenhouse gas (GHG) emissions from livestock exceed those of the transportation sector, accounting for 18% of global total emissions [4].

Currently, as the world's largest producer of livestock and poultry, China's livestock industry contributes to 80% of non-$CO_2$ GHG emissions from agricultural production activities [5]. The swine industry, as the core of China's livestock industry, plays an indispensable role in agricultural carbon reduction activities due to its large farming scale and high carbon emissions. According to data from the Food and Agriculture Organization of the United Nations, the livestock industry produces $7.1 \times 10^9$ tonnes of $CO_2$ annually, with pork production contributing about $6.39 \times 10^8$ tonnes [6]. Moreover, $CH_4$ and $N_2O$ emissions from pig farming in China account for 18% to 25% of the total emissions from livestock and poultry, making it the second-largest GHG emission source after cattle [7].

At present, China is the largest pork production and consumption market. According to data released by the U.S. Department of Agriculture, China's share of global pork production was 44.09% in 2021, ranking first globally. Furthermore, data from the Organization for Economic Co-operation and Development (OECD) and the U.S. Department of Agriculture indicate that China's pork consumption accounts for 46% of global pork consumption, with per capita pork consumption in China being twice the global average [8]. With the continuous growth in demand for meat products, the scale of China's swine industry is expanding. According to the National Bureau of Statistics, China's pig production increased by 27.4% in 2021 compared to the previous year, and pork production increased by 28.8%.

With the increase in pig production, the intestinal fermentation stage and manure management stage of pig farming release more $CH_4$ and $N_2O$. Additionally, the cultivation, transport, and processing of feed crops and the slaughtering of pigs also release more $N_2O$ and $CO_2$ due to the consumption of fertilizers and energy. With the introduction of the "Dual Carbon" goals, governments are gradually intensifying efforts to control livestock manure, and the standards for manure management are becoming increasingly stringent. This has increased pressure on pig farming enterprises and farms. Furthermore, due to the difficulties and high costs associated with grid connection for biogas power generation, some large-scale pig farms or enterprises are forced to release excess biogas directly into the atmosphere, exacerbating the growth of carbon emissions.

Although China's swine industry is gradually moving toward scale, intensification, and specialization, its industrialization level is still relatively low compared to developed countries with mature swine industries. Liu Xiaohong and Chen Yaosheng believe that with standardized management systems and advanced technologies, carbon emissions from the future livestock industry can be reduced by 30% [8]. It is evident that while the swine industry releases a large amount of greenhouse gases, it also holds significant potential for carbon reduction. Therefore, to achieve the low-carbon transformation of the swine

industry, the key lies in accurately understanding the current status of carbon emissions in the swine industry and its influencing factors.

### 1.2. Literature Review

In carbon emission estimation, a relatively comprehensive system for calculating carbon emissions has been established. Numerous scholars have researched the carbon emission situations in different countries. For example, Pao and Tsai used the IPCC emission factors to estimate carbon emissions from energy consumption in China, India, Brazil, and South Africa. China had the fastest growth rate in carbon emissions [9]. Zervas and Tsiplakou found that China's carbon emissions accounted for 21.18% of the total global carbon emissions [10].

As research progresses, scholars have gradually shifted their focus from a national level to a regional or industry level. The livestock industry, as a major source of global carbon emissions, has received widespread attention in academia. The mainstream methods for estimating carbon emissions from livestock include the emission factor method and the LCA method.

The emission factor method refers to the calculation of carbon emissions, focusing on the processes of animal intestinal fermentation and manure management, based on the carbon emission coefficients established by the Intergovernmental Panel on Climate Change (IPCC) or the Food and Agriculture Organization of the United Nations (FAO) [11]. For instance, Min Jisheng and Zhou Li used the emission factor method to estimate carbon emissions from 229 large-scale pig farms and found that collaboration between leading enterprises and pig farmers significantly reduced carbon emissions from medium- and large-scale pig farms [12]. Guo Dongsheng used the IPCC emission factors to estimate methane emissions from major livestock and poultry in China, highlighting that pig manure had the highest methane emissions, accounting for about 80% of total emissions from major livestock and poultry [13]. Hou Linge et al. studied a large-scale pig farm and found that the pig growth process is the primary contributor to carbon emissions in pig farms. Without considering the carbon dioxide produced by pig respiration, the carbon emissions from manure storage are the main determinant of carbon emissions in pig farms [14].

In contrast, the LCA method involves a more complex calculation, with relatively fewer research findings. This method assesses all inputs and outputs associated with the life cycle of a product or activity, from raw material acquisition to transportation, sales, use, recycling, and final disposal, evaluating all environmental impacts at each stage [11]. For example, Kong Fanbin et al. used the LCA method to analyze the spatiotemporal characteristics of carbon emissions from the swine industry in the Poyang Lake economic zone, identifying manure management and feed crop cultivation as the main sources of carbon emissions [15]. Zhou Jing et al. used the LCA method to estimate carbon emissions from pig farming in China. They found that large-scale pig farms had replaced free-range households as the main carbon emitters [7].

Differing from pig farming, the swine industry covers a much wider range. Based on the industry chain perspective, the swine industry includes feed production, pig farming, and slaughtering and processing.

However, the existing literature on carbon emissions from livestock based on the industry chain perspective mostly focuses on ruminants, such as dairy and beef cattle [16,17], lacking specific studies on estimating carbon emissions from the swine industry. Most of the literature often focuses on pig farming or pig farms in specific regions. The scope of carbon emission estimation is often limited to farming processes, neglecting carbon emissions from feed production or slaughtering and processing [7,14]. And these studies often emphasize the factors affecting carbon emissions at the pig farming stage, ignoring those at the feed production stage and the slaughtering and processing stage [12,18]. Therefore, the innovation of this paper is to focus on the swine industry from the perspective of the industry chain, covering feed production, pig farming, and slaughtering and processing. By using the LCA method and the LMDI decomposition model, this paper estimates the

carbon emissions from China's swine industry during 2001–2020, analyzes the dynamic evolution of the national swine industry's carbon emissions, studies the spatial differentiation of the provincial swine industries' carbon emissions, calculates the proportions of carbon emissions from each production stage of the swine industry, and decomposes the influencing factors of the swine industry's carbon emissions at a national level, provincial level, and production stage level.

## 2. Analysis of Carbon Emission Stages in the Swine Industry Based on the Life Cycle

Based on the LCA method [19], this paper estimates the carbon emissions of the swine industry at both the national and provincial levels for the years 2001–2020 from the perspective of the industry chain. The study involves a temporal characteristic analysis, spatial disparity analysis, and production stage contribution analysis of the estimated results. Furthermore, the LMDI decomposition model is applied to dissect the influencing factors of carbon emissions in the swine industry at the national, regional, and production stage levels. However, before calculating carbon emissions in the swine industry, it is essential to define core concepts such as the swine industry chain and carbon emissions in the swine industry and delineate the carbon emission stages in the swine industry chain.

### 2.1. Definition of Core Concepts

### 2.1.1. Swine Industry Chain

Before defining the concept of the swine industry chain, it is crucial to elucidate the definition of the swine industry. Scholars have different perspectives on the concept of the swine industry depending on their research focus. From a narrow perspective, some researchers [20–22] define the swine industry as the pig farming sector, primarily encompassing pig breeding and manure management. However, from a broad viewpoint, considering the industry chain, the swine industry includes the upstream feed crop cultivation and processing industry, the midstream pig farming industry, and the downstream pig slaughtering and processing industry, forming an industry chain starting from feed crop cultivation, based on pig farming, and ending with pig slaughtering and processing.

Therefore, from a broad perspective, this paper takes the swine industry chain as the research perspective, starting from the cultivation of feed crops to the end of pig slaughtering and processing, and divides the swine industry chain into six stages: feed crop cultivation, feed crop transportation and processing, intestinal fermentation, manure management, energy consumption in pig farming, and pig slaughter and processing.

### 2.1.2. Carbon Emissions in the Swine Industry

Carbon emissions in the swine industry, as referred to in this paper, are primarily limited to the carbon emissions generated in the production stages such as feed crop cultivation, feed crop transportation and processing, pig farming, and pig slaughter and processing. Therefore, in estimating carbon emissions from the swine industry, this paper calculates the carbon emissions for each of these six production stages: feed crop cultivation, feed crop transportation and processing, intestinal fermentation, manure management, energy consumption in pig farming, and pig slaughter and processing. The total carbon emissions in the swine industry are obtained by summing up the estimated emissions from these six production stages.

In calculating carbon emissions, scholars [14,21,22] commonly use the term "carbon dioxide equivalents ($CO_{2\text{-eq}}$)" to represent the emission levels for better understanding. For instance, when calculating carbon emissions from carbon dioxide ($CO_2$), methane ($CH_4$), and nitrous oxide ($N_2O$), scholars first calculate the GHG emissions such as $CO_2$, $CH_4$, and $N_2O$. Subsequently, $CH_4$ and $N_2O$ emissions are converted to $CO_2$ equivalent emissions by multiplying them with their respective global warming potential values. Finally, the carbon dioxide equivalent emissions of $CH_4$ and $N_2O$ are summed up with the emissions of $CO_2$, resulting in the overall carbon emissions. Since carbon emissions in the swine industry

are mainly made up of $CO_2$, $CH_4$, and $N_2O$, this study focuses exclusively on these three greenhouse gases in the carbon emissions calculation.

*2.2. Analysis of Carbon Emission Stages Based on the Life Cycle*

The swine industry referred to in this paper represents the broad concept of livestock production, including the pre-livestock plant cultivation layer, livestock production layer, and post-livestock processing layer [16].

Firstly, the pre-livestock plant production layer corresponds to the feed crop cultivation and processing industry. During the planting and processing of feed crops, production activities such as fertilization, the use of agricultural films, energy consumption, and transportation generate greenhouse gases. Within this stage, feed crop cultivation, transportation, and processing are the primary sources of carbon emissions.

Secondly, the livestock production layer corresponds to the pig farming industry. Livestock's normal metabolic activities, including microbial fermentation in the digestive tract, result in significant GHG emissions. Although ruminant animals are the main source of $CH_4$ emissions from intestinal fermentation [23], considering the large pig farming volume in China, $CH_4$ emissions from pig intestinal fermentation are also considered. Moreover, activities such as manure management and energy consumption in pig farming contribute substantially to carbon emissions.

Finally, the post-livestock processing layer corresponds to the pig slaughter industry. In this stage, pigs must undergo slaughtering, processing, and transportation before reaching consumers in a commodity form. These production activities also generate significant greenhouse gases due to energy consumption.

## 3. Research Methodology and Data Sources

*3.1. Estimation Method for Carbon Emissions in the Swine Industry*

3.1.1. Selection of Functional Unit

The selection of the functional unit is fundamental to carbon emission assessment, serving as the scale and measurement standard for mutual comparisons among research subjects [24]. This paper adopts the raising of one finished pig as the functional unit. As obtaining direct data on the annual raising quantity of pigs is challenging, this paper uses the annual output quantity of pigs as a basis. Following the approach of the Intergovernmental Panel on Climate Change [25], the annual raising quantity is calculated using the formula:

$$\text{AAP} = \frac{Day}{365} \times NA \tag{1}$$

where AAP represents the annual raising quantity in the swine industry, $Day$ is the average raising period for pigs, which is the average feeding days for pigs of different breeding scales, and $NA$ is the annual output quantity of pigs.

3.1.2. Emission Coefficients

In the existing literature, many researchers commonly use the GHG emission coefficients established by the IPCC in 2006. Although this calculation system underwent revisions and improvements in 2019, some data may still significantly differ from the current situation in China. To better align with the actual conditions of the swine industry in China, this paper primarily refers to the "Guidelines for Provincial Greenhouse Gas Inventory Compilation" published by the National Development and Reform Commission in 2011. Combining this with the research findings of domestic scholars, the carbon emission coefficients for each stage of the swine industry are determined (Table 1).

**Table 1.** Carbon Emission Coefficients for Different Stages in the Swine Industry.

| Symbol | Meaning | Emission Coefficient | Unit | Reference |
|---|---|---|---|---|
| | $CO_2$ equivalent emission coefficient during maize cultivation process | 0.7600 | $tCO_2$-eq/t | [26] |
| $ef_{u1}$ | $CO_2$ equivalent emission coefficient during soybean cultivation process | 0.1316 | $tCO_2$-eq/t | [27] |
| | $CO_2$ equivalent emission coefficient during wheat cultivation process | 0.5400 | $tCO_2$-eq/t | [26] |
| | $CO_2$ equivalent emission coefficient during maize transportation and processing | 0.0102 | $tCO_2$-eq/t | |
| $ef_{u2}$ | $CO_2$ equivalent emission coefficient during soybean transportation and processing | 0.1013 | $tCO_2$-eq/t | [28] |
| | $CO_2$ equivalent emission coefficient during wheat transportation and processing | 0.0319 | $tCO_2$-eq/t | |
| $ef_3$ | Methane ($CH_4$) emission coefficient from pig intestinal fermentation | 1.0000 | kg/head·y | |
| $ef_4$ | Methane ($CH_4$) emission coefficient from pig manure management | 3.4600 | kg/head·y | [23] |
| $ef_5$ | Nitrous oxide ($N_2O$) emission coefficient from pig manure management | 0.1970 | kg/head·y | |
| $ef_e$ | $CO_2$ emission coefficient from electricity consumption | 0.9944 | t/MW·h | |
| $ef_c$ | $CO_2$ emission coefficient from coal combustion | 1.9800 | t/t | [29] |
| $e$ | Thermal energy content of one unit of electricity | 0.0036 | KJ/MW·h | |
| | Proportion of maize in pig feed (average value) | 60.0000 | % | |
| $r_u$ | Proportion of soybean meal in pig feed (average value) | 20.0000 | % | [30,31] |
| | Proportion of wheat bran in pig feed (average value) | 12.5000 | % | |
| $p_u$ | By-product yield of soybean meal in soybean processing | 79.0000 | % | [32] |
| | By-product yield of wheat bran in wheat processing | 23.0000 | % | |
| $R_m$ | Energy coefficient of main product (soybean oil) in feed crops | 14.5100 | MJ/kg | |
| | Energy coefficient of main product (flour) in feed crops | 36.7700 | MJ/kg | [33] |
| $R_u$ | Energy coefficient of by-product (soybean meal) in feed crops | 13.8200 | MJ/kg | |
| | Energy coefficient of by-product (wheat bran) in feed crops | 4.1500 | MJ/kg | |
| $GHP_1$ | Global warming potential of $CH_4$ | 21.0000 | — | [34] |
| $GHP_2$ | Global warming potential of $N_2O$ | 310.000 | — | |
| $MJ$ | Slaughter and processing energy consumption per kg of pork product | 3.7600 | KJ/kg | [35] |

### 3.1.3. Estimation Formula for Carbon Emissions in the Swine Industry

With reference to the "Guidelines for Provincial Greenhouse Gas Inventory Compilation" and existing literature [15,36], the following estimation formula for carbon emissions in the swine industry is constructed.

(1) Feed Crop Cultivation Stage: In China, pig farming feed mainly comprises seven ingredients: corn, soybean meal, wheat bran, calcium hydrogen phosphate, salt, trace elements, and compound vitamins. Corn accounts for 55–65% of the feed ingredients, soybean meal around 15–25%, and wheat bran about 10–15%. Calcium hydrogen phosphate constitutes only 2%, while salt, trace elements, and compound vitamins comprise less than 0.5% [30,31]. Since the carbon emissions from ingredients like calcium hydrogen phosphate and salt are negligible, and it is more difficult to obtain certain data, this paper calculates carbon emissions only from the cultivation of corn, soybeans, and wheat. Additionally, as soybean meal and wheat bran in pig feed are by-products of soybean and wheat processing, the carbon emissions from the cultivation of soybeans and wheat cannot be entirely attributed to soybean meal and wheat bran. Therefore, this paper uses the energy allocation method to assign carbon emissions from soybean and wheat cultivation to their respective products: soybean meal and oil, wheat bran, and flour. The coefficient for corn is one, as it requires no allocation. The specific calculation formula is as follows:

$$Ecp = \Sigma AAP \times m \times r_u \div p_u \times ef_{u1} \times F_u \tag{2}$$

$$F_u = \frac{R_u \times P_u}{R_u \times P_u + R_m \times (1 - P_u)} \tag{3}$$

(2) Feed Crop Transportation and Processing Stage: The feed ingredients produced, such as corn, soybean meal, and wheat bran, need to undergo various processes, including drying, grinding, batching, mixing, transportation, and processing, to become pig feed. The energy consumed in this process generates significant GHG emissions. The formula is as follows:

$$Egp = \Sigma AAP \times m \times r_u \div p_u \times ef_{u2} \times F_u \tag{4}$$

(3) Intestinal Fermentation Stage: $CH_4$ emissions from the intestinal fermentation stage are a major source of carbon emissions in the swine industry. In the anaerobic environment of the digestive system, pigs produce a significant amount of $CH_4$. This paper adopts the

default emission factor recommended in the "Guidelines for Provincial Greenhouse Gas Inventory Compilation" to calculate $CH_4$ emissions from pig intestinal fermentation. The calculation formula is as follows:

$$\text{Eef} = \text{AAP} \times ef_3 \times GHP_1 \tag{5}$$

(4) Manure Management Stage: The GHG emissions from pig manure management depend on the storage or management method. When manure is stored in liquid form, it undergoes anaerobic degradation, producing $CH_4$. When manure is treated in solid form, it undergoes aerobic degradation, producing $N_2O$ through the processes of nitrification and denitrification [15]. The calculation formula for emissions in this stage is as follows:

$$\text{Emm} = \text{AAP} \times ef_4 \times GHP_1 + \text{AAP} \times ef_5 \times GHP_2 \tag{6}$$

(5) Energy Consumption in Pig Farming Stage: In this stage, various mechanical transport equipment, environmental control equipment such as heating lamps and fans, and other facilities like coal and boilers consume diesel, gasoline, and electricity produce greenhouse gases during operation. Following the approach of Kong Fanbin et al., this paper calculates the carbon emissions from energy consumption in pig farming based on the annual raising quantity [15].

$$\text{Edh} = \text{AAP} \times \frac{cost_e}{price_e} \times ef_e + \text{AAP} \times \frac{cost_c}{price_c} \times ef_c \tag{7}$$

(6) Slaughter and Processing Stage: Before pigs are sold in commodity form, they need to be transported from the farm to the slaughterhouse. After undergoing processes like slaughtering, sterilization, packaging, and transportation, they become pork products in the market. This stage also generates GHG emissions due to energy consumption. The specific calculation formula is as follows:

$$\text{Emp} = \text{Q} \times \frac{MJ}{e} \times ef_e \tag{8}$$

(7) Total Carbon Emissions in the Swine Industry: Based on the carbon emissions of the six production stages mentioned above, the formula for calculating the total carbon emissions in the swine industry in China is as follows:

$$\text{Etotal} = \text{Ecp} + \text{Egp} + \text{Eef} + \text{Emm} + \text{Edh} + \text{Emp} \tag{9}$$

In the above calculation formulas, except for the carbon emission coefficients listed in Table 1, the meanings and units of the remaining symbols are shown in Table 2.

**Table 2.** Meanings and Units of Symbols in the Carbon Emission Estimation Formula of the Swine Industry.

| Symbol | Meaning | Unit |
|---|---|---|
| Ecp | $CO_2$ emissions from the feed crop cultivation stage | $10^4$ tonnes |
| Egp | $CO_2$ emissions from the feed crop transportation and processing stage | $10^4$ tonnes |
| Eef | $CO_2$ equivalent emissions from the intestinal fermentation stage | $10^4$ tonnes |
| Emm | $CO_2$ equivalent emissions from the manure management stage | $10^4$ tonnes |
| Edh | $CO_2$ emissions from the energy consumption in pig farming stage | $10^4$ tonnes |
| Emp | $CO_2$ emissions from the pig slaughter and processing stage | $10^4$ tonnes |
| Etotal | Total carbon emissions from the swine industry | $10^4$ tonnes |
| AAP | Annual raising quantity of pigs | $10^4$ heads |
| *NA* | Annual output quantity of pigs | $10^4$ heads |
| *Day* | Average raising cycle of pigs, taking the average days of pig raising for different farming scales | Days |
| Q | Annual production quantity of pork products | $10^4$ tonnes |
| m | Average consumption quantity of concentrate feed per pig, taking the average consumption for different pig scales | Tonnes/head |
| $F_u$ | Carbon emission distribution coefficient for feed ingredient *u*, where soybean meal is 78.18%, wheat bran is 3.26%, and corn is 100% as it does not need distribution. | % |

| Symbol | Meaning | Unit |
|---|---|---|
| $cost_e$ | Average electricity cost per pig in pig farming | CNY/head |
| $cost_c$ | Average coal cost per pig in pig farming | CNY/head |
| $price_e$ | Average unit electricity price in pig farming | CNY/kWh |
| $price_c$ | Average unit coal price in pig farming | CNY/ton |

*3.2. Decomposition Method of Carbon Emissions Factors in the Swine Industry*

Understanding the influencing factors and their effects on carbon emissions is essential for carbon reduction strategies for the swine industry. Common methods for decomposing the influencing factors of carbon emission include the IPAT model, STIRPAT model, and Kaya model [37]. Among them, the Kaya model, which utilizes principles of calculus, decomposes carbon emissions from human social activities into four factors: population, residents' living standards, energy intensity, and carbon emissions. Due to its operational and controllable characteristics, the Kaya model has been widely applied in research on carbon emissions factors in different countries or regions.

However, when using calculus principles to calculate the influencing factors of carbon emission, residual terms often appear, leading to significant errors in the model's decomposition results. The LMDI decomposition model can effectively eliminate residual terms, overcoming the shortcomings of other models [38].

Therefore, taking into account the characteristics of the above decomposition methods and considering the complexity of the influencing factors of carbon emissions in the swine industry and the accessibility of data, this paper, based on the Kaya model, employs the LMDI decomposition model to quantitatively analyze the factors influencing carbon emissions in the swine industry.

Following the basic form of the Kaya model, this paper draws on existing research findings [22,39] to appropriately modify the Kaya model, decomposing carbon emissions in China's swine industry into five factors: production efficiency, industry structure, economic development, urbanization level, and population size. The specific expressions are as follows:

$$C = \frac{C}{GDP_P} \times \frac{GDP_P}{GDP_A} \times \frac{GDP_A}{P_R} \times \frac{P_R}{P_T} \times P_T \tag{10}$$

$$C = EI \times AI \times CI \times UR \times P_T \tag{11}$$

In these equations, $C$ represents the total carbon emissions in the swine industry; $GDP_P$ is the total output value of pig farming; $GDP_A$ is the total output value of the livestock industry; $P_R$ is the total rural population; $P_T$ is the total population; $EI = C/GDP_P$ is the ratio of carbon emissions in the swine industry to the total output value of pig farming, representing the carbon emission per unit output value in pig farming and indicating the impact of changes in production efficiency on carbon emissions; $AI = GDP_P/GDP_A$ is the proportion of the total output value of pig farming to the total output value of the livestock industry, indicating the impact of changes in industry structure on carbon emissions in the swine industry; $CI = GDP_A/P_R$ is the ratio of the total output value of the livestock industry to the total rural population, reflecting the economic development level of the local rural area; and $UR = P_R/P_T$ is the ratio of the total rural population to the total population, reflecting the level of urbanization in the local area.

In the LMDI decomposition model, the results of the 'additive decomposition model' and the 'multiplicative decomposition model' are consistent, and the factors revealed by the 'additive decomposition model' are clear [22]. Therefore, this paper chooses the LMDI additive decomposition model to further decompose Equation (11) for quantifying the impact of each factor on carbon emissions. The specific expressions are as follows:

$$\Delta C = C_t - C_0 = \Delta EI + \Delta AI + \Delta CI + \Delta UR + \Delta P_T \tag{12}$$

$$\Delta EI = \sum \frac{C_t - C_0}{lnC_t - lnC_0} \times (lnEI_t - lnEI_0) \tag{13}$$

$$\Delta AI = \sum \frac{C_t - C_0}{lnC_t - lnC_0} \times (lnAI_t - lnAI_0) \tag{14}$$

$$\Delta CI = \sum \frac{C_t - C_0}{lnC_t - lnC_0} \times (lnCI_t - lnCI_0) \tag{15}$$

$$\Delta UR = \sum \frac{C_t - C_0}{lnC_t - lnC_0} \times (lnUR_t - lnUR_0) \tag{16}$$

$$\Delta P_T = \sum \frac{C_t - C_0}{lnC_t - lnC_0} \times (lnP_{Tt} - lnP_{T0}) \tag{17}$$

In the above model, t represents the period ($t = 1, 2, 3, \ldots, n$), and $t_0$ represents the base period. $\Delta EI$, $\Delta AI$, $\Delta CI$, $\Delta UR$, and $\Delta P_T$ represent the contribution of changes in production efficiency, industry structure, economic development, urbanization level, and population size to the carbon emissions of the swine industry, respectively, from the base period to period t. The cumulative contribution for each influencing factor over the study period is the sum of the contribution values for each year, expressed in units of $10^4$ tonnes.

*3.3. Data Sources*

This paper mainly involves four aspects of original data: Firstly, data on the annual output quantity of pigs and the annual production quantity of pork in China and various provinces and municipalities are sourced from the "China Animal Husbandry and Veterinary Yearbook 2002–2021 (China Animal Husbandry and Veterinary Yearbook Editorial Committee. China Animal Husbandry and Veterinary Yearbook [DB/OL] https://data.cnki.net/trade/yearBook/single?zcode=Z009&id=N2023030190, accessed on 18 June 2023)".

Secondly, data on electricity and coal expenses, feed consumption per pig, and average raising days under different breeding models in various provinces and municipalities across the country primarily come from the "Compilation of National Agricultural Cost-Benefit Data 2002–2021 (Prices Department of NDRC. Compilation of National Agricultural Cost–Benefit Data [DB/OL] https://www.zgtjnj.org/navibooklist-n3023062107-1.html, accessed on 18 June 2023)". Additionally, due to the unavailability of direct data on electricity and coal consumption in pig farming at the national and regional levels, this paper calculates the electricity and coal expenses per pig under different breeding models by dividing them by the corresponding electricity and coal prices.

Thirdly, the data on electricity prices for agriculture are obtained from the "Sales Tariff Table" published on the official websites of the State Grid Corporation (State grid. State Grid Sales Tariffs for Residential and Agricultural Electricity [DB/OL] https://www.95598.cn/osgweb/ipElectrovalenceStandard, accessed on 23 September 2023). The average electricity prices are taken for different voltage levels. Coal prices are based on the average monthly market prices of anthracite coal from the official websites of the PRC National Development and Reform Commission (NDRC. Changes in Market Prices of Important Means of Production in Circulation [DB/OL] https://www.ndrc.gov.cn/fggz/cyfz/zcyfz/202108/t20210825_1294520.html, accessed on 23 September 2023) and the "China Price Statistical Yearbook 2021 (Urban Socio-Economic Surveys Division, National Statistical Office. China Price Statistical Yearbook [DB/OL] https://data.cnki.net/yearBook/single?id=N2021100073, accessed on 18 June 2023)". For years with missing data, this paper uses the anthracite coal ex-factory price index to estimate the missing market prices.

Fourthly, the total output value of pig farming and animal husbandry, along with the rural population, is sourced from the "China Rural Statistics Yearbook 2002–2021 (Rural Socio-Economic Surveys Division of National Statistical Office. China Rural Statistics Yearbook [DB/OL] https://data.cnki.net/trade/yearBook/single?zcode=Z009&id=N2024010048, accessed on 18 June 2023)". The total population at the national and provincial levels is sourced from the "China Statistical Yearbook 2002–2021 (PRC National Bureau of Statistics. China Statistical Yearbook [DB/OL] https://data.cnki.net/trade/yearBook/single?zcode=Z009&id=N2023110024, accessed on 18 June 2023)". Considering that GDP

lacks vertical comparability, the total output value of pig farming and animal husbandry is adjusted to comparable actual values based on the benchmark year 2001.

## 4. Results and Analysis of Carbon Emissions in the Swine Industry

*4.1. Analysis of the Dynamic Evolution of Carbon Emissions in the National Swine Industry*

4.1.1. Overall Trends in Carbon Emissions in the National Swine Industry

Based on the fluctuation trend, the total carbon emissions in China's swine industry experienced "slow growth—rapid decline—fluctuation increase—fluctuation decrease" from 2001 to 2020, with significant retreats in 2007 and 2019 (Table 3). The highest value was 101.08 million tonnes in 2014, and the lowest was 75.07 million tonnes in 2007, showing noticeable fluctuations. In addition, the changes in carbon emissions in the swine industry correspond to the increase or decrease in pig production (Table 4), indicating that the growth in demand for pork consumption, which drives an increase in pig supply, is the root cause of the rise in carbon emissions in the swine industry. The improvement of people's material living standards comes at the expense of the environment [7].

**Table 3.** Carbon Emissions from the Swine Industry in China (2001–2020, in $10^4$ tonnes $CO_2$-eq).

| Year | Feed Crop Cultivation | Feed Crop Transportation and Processing | Intestinal Fermentation | Manure Management | Energy Consumption in Pig Farming | Pig Slaughter and Processing | Total |
|---|---|---|---|---|---|---|---|
| 2001 | 3081.54 | 167.58 | 521.52 | 3321.11 | 723.59 | 4.25 | 7819.60 |
| 2002 | 3086.25 | 167.84 | 531.59 | 3385.20 | 538.58 | 4.40 | 7713.85 |
| 2003 | 3375.77 | 183.58 | 558.59 | 3557.17 | 546.59 | 4.59 | 8226.30 |
| 2004 | 3541.01 | 192.57 | 584.31 | 3720.96 | 557.32 | 4.78 | 8600.94 |
| 2005 | 3681.52 | 200.21 | 601.50 | 3830.39 | 391.12 | 5.09 | 8709.83 |
| 2006 | 3814.30 | 207.43 | 625.78 | 3985.05 | 481.97 | 5.28 | 9119.82 |
| 2007 | 3234.06 | 175.87 | 507.18 | 3229.78 | 355.92 | 4.36 | 7507.18 |
| 2008 | 3602.56 | 195.91 | 540.62 | 3442.74 | 300.39 | 4.70 | 8086.94 |
| 2009 | 3802.79 | 206.80 | 565.27 | 3599.69 | 293.91 | 4.97 | 8473.43 |
| 2010 | 3831.19 | 208.35 | 576.86 | 3673.51 | 293.82 | 5.16 | 8588.89 |
| 2011 | 3929.14 | 213.67 | 576.60 | 3671.87 | 298.01 | 5.14 | 8694.44 |
| 2012 | 4251.57 | 231.21 | 612.84 | 3902.62 | 300.11 | 5.43 | 9303.78 |
| 2013 | 4484.49 | 243.87 | 632.52 | 4027.96 | 301.38 | 5.58 | 9695.81 |
| 2014 | 4705.92 | 255.92 | 654.61 | 4168.65 | 316.93 | 5.77 | 10,107.79 |
| 2015 | 4463.10 | 242.71 | 620.54 | 3951.66 | 295.98 | 5.58 | 9579.57 |
| 2016 | 4474.62 | 243.34 | 608.65 | 3875.93 | 301.50 | 5.39 | 9509.42 |
| 2017 | 4695.85 | 255.37 | 631.92 | 4024.12 | 293.16 | 5.54 | 9905.96 |
| 2018 | 4766.26 | 259.20 | 632.23 | 4026.11 | 294.81 | 5.49 | 9984.10 |
| 2019 | 3632.92 | 197.56 | 489.64 | 3118.08 | 237.28 | 4.33 | 7679.81 |
| 2020 | 3765.84 | 204.79 | 485.65 | 3092.64 | 253.83 | 4.18 | 7806.93 |

**Table 4.** Historical averages of original data on the national swine industry (2001–2020).

| Year | Output of Pigs $10^4$ Heads | Pork Output $10^4$ Tonnes | Feeding Days Days | Electricity CNY/Head | Coal Cost CNY/Head | Concentrate Feed Consumption Tonnes/Head |
|---|---|---|---|---|---|---|
| 2001 | 54,936.80 | 4184.50 | 165.00 | 2.38 | 4.69 | 252.40 |
| 2002 | 56,684.00 | 4326.60 | 163.00 | 2.17 | 3.89 | 248.00 |
| 2003 | 59,200.49 | 4518.61 | 164.00 | 2.24 | 3.83 | 258.15 |
| 2004 | 61,800.70 | 4701.61 | 164.33 | 2.58 | 4.20 | 258.87 |
| 2005 | 66,098.60 | 5010.61 | 158.17 | 2.54 | 3.15 | 261.45 |
| 2006 | 68,050.36 | 5197.17 | 159.83 | 3.33 | 3.66 | 260.37 |
| 2007 | 56,508.27 | 4287.82 | 156.00 | 3.10 | 3.40 | 272.38 |
| 2008 | 61,016.60 | 4620.50 | 154.00 | 3.07 | 2.83 | 284.65 |
| 2009 | 64,538.60 | 4890.80 | 152.23 | 3.17 | 2.41 | 287.37 |
| 2010 | 66,686.43 | 5071.24 | 150.35 | 3.18 | 2.52 | 283.70 |
| 2011 | 66,170.31 | 5053.13 | 151.46 | 3.36 | 2.74 | 291.08 |
| 2012 | 69,789.50 | 5342.70 | 152.63 | 3.50 | 2.12 | 296.35 |
| 2013 | 71,557.30 | 5493.00 | 153.64 | 3.58 | 1.65 | 302.85 |
| 2014 | 73,510.40 | 5671.40 | 154.78 | 3.63 | 1.50 | 307.08 |
| 2015 | 70,825.00 | 5486.50 | 152.29 | 3.58 | 1.31 | 307.23 |
| 2016 | 68,502.00 | 5299.10 | 154.43 | 3.61 | 1.37 | 314.04 |
| 2017 | 70,202.10 | 5451.80 | 156.45 | 3.90 | 0.99 | 317.43 |
| 2018 | 69,382.40 | 5403.70 | 158.38 | 3.94 | 1.20 | 322.03 |
| 2019 | 54,419.20 | 4255.30 | 156.39 | 4.13 | 1.08 | 316.94 |
| 2020 | 52,704.10 | 4113.30 | 160.16 | 4.41 | 1.03 | 331.24 |
| Mean | 64,129.16 | 4918.97 | 156.88 | 3.27 | 2.48 | 288.68 |

### 4.1.2. Slow Growth Phase (2001–2006)

During this period, the total carbon emissions from China's swine industry gradually increased from 78.20 million tonnes to 91.20 million tonnes, with an average annual growth rate of 3.16%. The main reasons can be attributed to the structural adjustment of the swine industry and the influence of agricultural policy orientation.

To increase farmers' income and adjust the agricultural structure, China's swine industry shifted from pursuing quantity growth to emphasizing quality and efficiency improvement since the end of the last century. Pig farming evolved from free-range raising to a more specialized and scaled-up approach. The focus of industry development gradually tilted toward advantageous regions, forming a preliminary structure where pig enterprises drove small-scale farmers [40]. This accelerated the pace of integration in the swine industry, promoting its overall development. Moreover, in 2001, China's Ministry of Agriculture issued the "Opinions on Accelerating the Development of Animal Husbandry", advocating for the vigorous development of animal husbandry to drive related industries such as planting and reducing taxes on animal husbandry and slaughter [41]. These measures significantly boosted the enthusiasm of farmers involved in pig and poultry farming, fostering the growth of China's swine industry and increasing carbon emissions.

### 4.1.3. Sharp Decline Phase (2006–2007)

During this period, the total carbon emissions from the swine industry dropped sharply from 91.20 million tonnes to 75.07 million tonnes, a substantial decrease of 17.68%. The primary reason was the outbreak of pig diseases and the lack of confidence among farmers in the market. In the first half of 2006, pig market prices remained persistently low, causing significant losses for many farmers. Although the market improved in the second half of the year, widespread highly pathogenic diseases caused significant pig deaths, leaving many farmers helpless [42]. These challenges eroded pig farmers' confidence and dampened their enthusiasm to resume stockpiling. Additionally, the simultaneous increase in pig prices and rising pig farming costs in 2007 directly led to a 16.96% decrease in China's pig output compared to 2006, causing a substantial reduction in carbon emissions in the swine industry.

### 4.1.4. Fluctuating Increase Phase (2007–2018)

Despite minor reductions in carbon emissions in 2015 and 2016, the overall trend increased. Compared to 2007, the carbon emissions from the swine industry increased by 33% in 2018, with an average annual growth rate of about 2.69%. This continuous growth is attributed to the increased proportion of large-scale pig farming and innovations in pig farming models.

Since 2007, the central government has allocated special funds to support standardized and large-scale pig farming. This targeted support is specifically directed to farms or communities with an annual output exceeding 500 pigs. Meanwhile, various new farming models emerged, fostering a trend of vertical and horizontal integration. On the one hand, some pig slaughter and meat processing enterprises extended upstream into the pig farming sector to ensure material safety, build brand image, and reduce market risks, such as Shuanghui and COFCO Joycome. On the other hand, upstream feed or farming enterprises extended downstream into the farming or slaughtering processes, like New Hope Liuhe and Chuying Agriculture. Additionally, certain companies collaborated with free-range farmers through models like "company + farmer" or "company + third-party manager + farmer," providing technical guidance and supplies such as piglets, feed, and veterinary drugs to enhance the risk resilience and efficiency of free-range farming [43]. These measures have elevated the proportion of large-scale pig farming, promoted the transformation and development of pig farming, and increased the carbon emissions of the swine industry.

However, from 2015 to 2016, the pig market supply declined. In 2016, pig production and pork output decreased by 6.8% and 6.6%, respectively, compared to 2014. The main

reasons included sustained losses in pig farming profitability from 2014 to 2015, lower farming efficiency, and the enforcement of the "Regulations on the Prevention and Control of Pollution from Large-scale Livestock and Poultry Farming" in various regions from 2015 to 2016. This led to intensified pollution control efforts, forcing many environmentally non-compliant small and medium-sized farms or farming enterprises to shut down, resulting in a significant reduction in pig production capacity.

4.1.5. Fluctuating Decrease Phase (2018–2020)

During this period, carbon emissions from the swine industry experienced a 26.12% decrease in 2019 compared to the previous year, but a slight rebound of 1.66% occurred in 2020. The three main reasons for these fluctuations are as follows: First, the outbreak of African swine fever in 2018 had a significant impact on the pig market. The culling of pigs in affected areas, restrictions on pig transportation, and a rapid increase in piglet prices made pig farming increasingly challenging, leading to a significant nationwide decline in the pig population and the number of breeding sows. Second, the implementation of the "Environmental Protection Tax Law" in January 2018 imposed environmental taxes on farms with livestock holdings exceeding 50 cattle, 500 pigs, or 5000 chickens and ducks, playing a role in curbing carbon emissions in the swine industry [41]. Third, by 2020, China's pig population had recovered to over 90% of the normal year (2017), with 406.5 million pigs and 41.6 million breeding sows. However, the development of China's swine industry was impacted due to the shortage of breeding pigs and the preservation of breeding sows caused by African swine fever. Moreover, the outbreak of the COVID-19 pandemic in 2019 also affected the production and distribution of pigs, forcing a slowdown in the recovery pace of swine industry capacity. Pig production and pork output continued to decline in 2020, and adjusting the pig farming structure and quickly restoring pig production faced significant pressure in the short term [44].

*4.2. Analysis of the Spatial Characteristics of Carbon Emissions in the Swine Industry at the Provincial Level*

4.2.1. Spatial Characteristics of Carbon Emissions in the Swine Industry at the Provincial Level

The spatial differentiation of carbon emissions from the provincial swine industries is significant due to the large geographical differences in the original data of China's swine industry (Table 5). In 2020, the total carbon emissions from China's swine industry were 78.07 million tonnes, with Sichuan, Hunan, Henan, Yunnan, Shandong, Hebei, Hubei, Guangdong, Guangxi, and Jiangxi ranking among the top ten provinces, each exceeding 3 million tonnes (Table 6).

Due to significant regional differences in carbon emissions from the swine industry, this study employs carbon intensity to clearly assess the provincial spatial differentiation of carbon emissions in the swine industry. Carbon intensity measures the carbon emissions produced per head of output pig and represents the local swine industry's carbon emission reduction level. As shown in Table 6, the average carbon intensity from China's swine industry in 2020 was 0.3409 tonnes/head. Xinjiang had the highest carbon intensity (0.38 tonnes/head), while Yunnan had the lowest (0.332 tonnes/head). The top ten regions in carbon intensity were Xinjiang, Inner Mongolia, Shanghai, Fujian, Qinghai, Beijing, Anhui, Guizhou, Henan, and Jiangsu, all of which exceeded the national average.

**Table 5.** Historical averages of original data on the provincial swine industries (2001–2020).

| Region | Output of Pigs<br>$10^4$ Heads | Pork Output<br>$10^4$ Tonnes | Electricity<br>Yuan/Head | Coal Cost<br>Yuan/Head | Concentrate Feed Consumption<br>Tonnes/Head |
|---|---|---|---|---|---|
| Beijing | 307.91 | 23.02 | 9.89 | 4.76 | 280.53 |
| Tianjin | 348.93 | 26.67 | 4.57 | 1.29 | 287.26 |
| Hebei | 3562.48 | 271.40 | 2.75 | 0.34 | 265.23 |
| Shanxi | 695.56 | 53.59 | 2.92 | 3.06 | 309.93 |
| Inner Mongolia | 881.64 | 72.94 | 5.57 | 6.96 | 344.50 |
| Liaoning | 2362.28 | 199.74 | 3.12 | 0.60 | 336.08 |
| Jilin | 1405.30 | 112.90 | 2.37 | 0.64 | 328.03 |
| Heilongjiang | 1574.97 | 116.73 | 2.84 | 1.71 | 303.35 |
| Shanghai | 245.85 | 16.53 | 8.80 | 0.10 | 250.16 |
| Jiangsu | 2777.98 | 208.21 | 3.05 | 0.66 | 258.45 |
| Zhejiang | 1576.42 | 111.33 | 4.93 | 0.48 | 289.03 |
| Anhui | 2663.64 | 226.10 | 3.62 | 1.90 | 285.44 |
| Fujian | 1728.56 | 130.15 | 5.88 | 1.03 | 292.83 |
| Jiangxi | 2637.84 | 206.52 | 4.45 | 2.20 | 306.47 |
| Shandong | 4293.42 | 351.73 | 2.28 | 1.13 | 262.96 |
| Henan | 5357.17 | 411.32 | 5.07 | 1.73 | 282.53 |
| Hubei | 3626.37 | 276.44 | 3.17 | 0.48 | 314.16 |
| Hunan | 5698.30 | 411.80 | 2.22 | 3.22 | 322.49 |
| Guangdong | 3456.88 | 253.73 | 4.50 | 4.62 | 270.11 |
| Guangxi | 3007.68 | 212.29 | 2.97 | 0.79 | 257.57 |
| Hainan | 456.65 | 37.63 | 2.54 | 2.30 | 262.80 |
| Chongqing | 1887.30 | 138.65 | 4.36 | 7.79 | 304.12 |
| Sichuan | 6639.21 | 471.78 | 3.31 | 4.37 | 245.73 |
| Guizhou | 1609.94 | 143.49 | 3.44 | 5.90 | 273.91 |
| Yunnan | 3017.15 | 254.49 | 2.50 | 3.68 | 299.53 |
| Shaanxi | 1029.21 | 76.80 | 2.49 | 2.94 | 288.95 |
| Gansu | 663.64 | 48.13 | 2.64 | 1.99 | 290.30 |
| Qinghai | 117.90 | 8.55 | 2.07 | 3.96 | 275.79 |
| Ningxia | 119.53 | 8.54 | 1.63 | 0.99 | 286.61 |
| Xinjiang | 363.31 | 27.14 | 4.16 | 5.79 | 285.12 |

**Table 6.** Total Carbon Emissions and Carbon Intensity in China's Provinces and Municipalities (2020, in $10^4$ tonnes $CO_2$-eq and tonnes/head).

| Region | Total Carbon Emissions | Carbon Intensity | Region | Total Carbon Emissions | Carbon Intensity |
|---|---|---|---|---|---|
| Beijing | 2.68 | 0.3476 | Henan | 648.15 | 0.3426 |
| Tianjin | 28.98 | 0.3405 | Hubei | 384.87 | 0.3334 |
| Hebei | 423.71 | 0.3321 | Hunan | 682.05 | 0.3336 |
| Shanxi | 118.78 | 0.3394 | Guangdong | 372.85 | 0.3349 |
| Inner Mongolia | 116.24 | 0.3570 | Guangxi | 338.26 | 0.3379 |
| Liaoning | 319.42 | 0.3347 | Hainan | 38.22 | 0.3322 |
| Jilin | 192.82 | 0.3325 | Chongqing | 210.38 | 0.3342 |
| Heilongjiang | 264.07 | 0.3362 | Sichuan | 835.85 | 0.3393 |
| Shanghai | 15.18 | 0.3540 | Guizhou | 250.42 | 0.3434 |
| Jiangsu | 274.03 | 0.3421 | Yunnan | 503.03 | 0.3320 |
| Zhejiang | 99.72 | 0.3415 | Shaanxi | 144.20 | 0.3337 |
| Anhui | 327.87 | 0.3475 | Gansu | 99.11 | 0.3400 |
| Fujian | 199.94 | 0.3505 | Qinghai | 6.86 | 0.3483 |
| Jiangxi | 328.23 | 0.3372 | Ningxia | 14.53 | 0.3358 |
| Shandong | 489.88 | 0.3338 | Xinjiang | 83.94 | 0.3800 |

### 4.2.2. Classification of China's Provincial Regions Based on the Swine Industry's Carbon Emissions

Using the national averages of total carbon emissions and carbon intensity in China's swine industry in 2020 as classification criteria, the 30 provinces and municipalities are divided into four types, as illustrated in Figure 1.

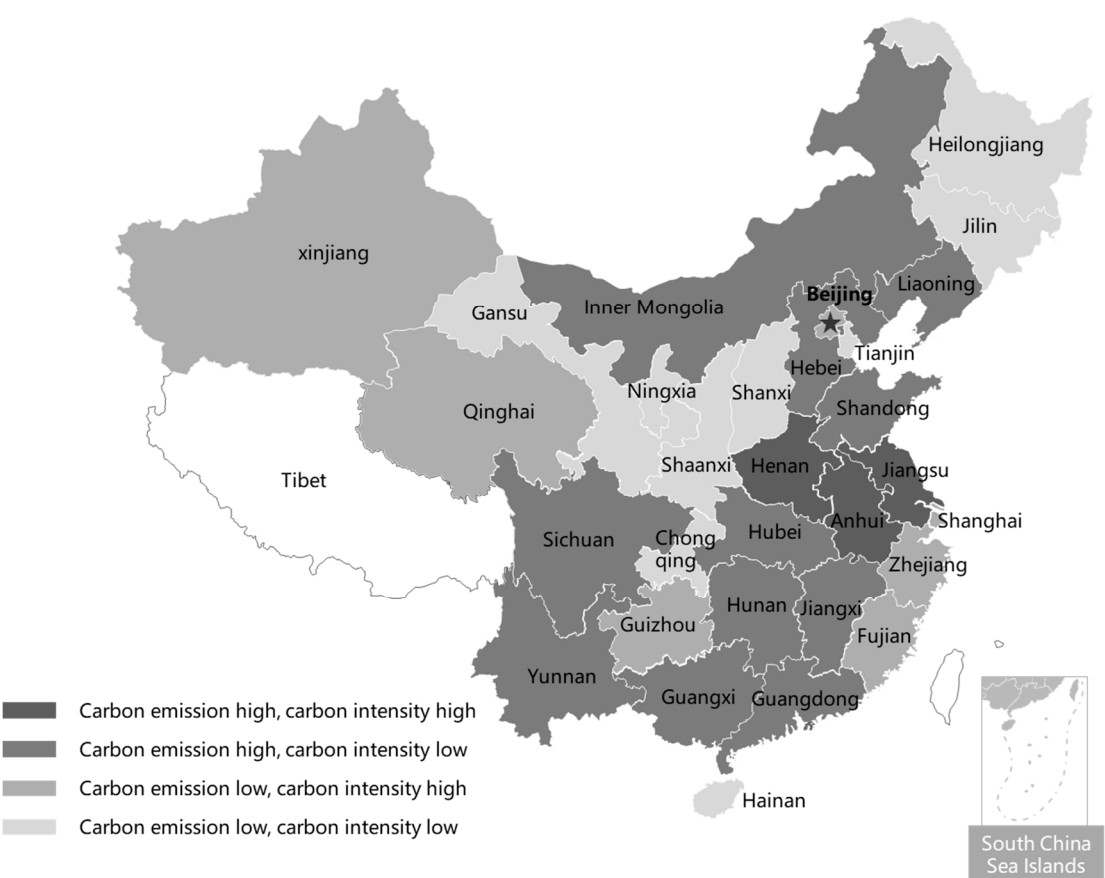

**Figure 1.** Classification of China's Provincial Regions Based on Total Carbon Emissions and Unit Carbon Emissions of the Swine Industry.

The first type comprises provinces with both high total carbon emissions and high carbon intensity, including Jiangsu, Anhui, and Henan. These regions prioritize the economic benefits of the swine industry while neglecting carbon emissions. These regions should adjust the direction of industrial development and focus on controlling the total carbon emissions and carbon intensity of the swine industry.

The second type includes provinces with high total carbon emissions and low carbon intensity, such as Hebei, Liaoning, Inner Mongolia, Jiangxi, Shandong, Hubei, Hunan, Guangdong, Guangxi, Sichuan, and Yunnan. These regions have performed relatively well in carbon emission reduction in the swine industry and can be considered key planning areas for the low-carbon development of China's swine industry.

The third type encompasses provinces with low total carbon emissions and high carbon intensity, including Beijing, Shanghai, Zhejiang, Fujian, Guizhou, Qinghai, and Xinjiang. Due to the lower economic output of the swine industry in these areas and insufficient government attention to carbon emissions, the carbon intensity is higher. Therefore, these provinces should focus on achieving technological breakthroughs in carbon reduction for the swine industry.

The fourth type consists of provinces with both low total carbon emissions and low carbon intensity, including Tianjin, Shanxi, Jilin, Heilongjiang, Hainan, Chongqing, Shaanxi, Gansu, and Ningxia. In these regions, either the swine industry is not a local focal point, or the technological proficiency in the swine industry is high.

*4.3. Analysis of Carbon Emission Contributions in Production Stages of the Swine Industry*

4.3.1. Analysis of Carbon Emission Contributions in Six Production Stages

As shown in Table 7, the average proportion of carbon emissions from each production stage in the total carbon emissions of the swine industry from 2001 to 2020 is as follows, in

descending order: feed crop cultivation (44.50%), manure management (42.19%), intestinal fermentation (6.63%), energy consumption in pig farming (4.30%), feed crop transportation and processing (2.32%), and slaughter and processing (0.06%). The feed crop cultivation stage and manure management stage are the primary contributors to carbon emissions in the swine industry.

**Table 7.** Proportion of Carbon Emissions from Various Production Stages in China's Swine Industry (2001–2020, in %).

| Year | Feed Crop Cultivation | Manure Management | Intestinal Fermentation | Energy Consumption in Pig Farming | Feed Crop Transportation and Processing | Pig Slaughter and Processing |
|---|---|---|---|---|---|---|
| 2001 | 39.41 | 42.47 | 6.67 | 9.25 | 2.14 | 0.05 |
| 2002 | 40.01 | 43.88 | 6.89 | 6.98 | 2.18 | 0.06 |
| 2003 | 41.04 | 43.24 | 6.79 | 6.64 | 2.23 | 0.06 |
| 2004 | 41.17 | 43.26 | 6.79 | 6.48 | 2.24 | 0.06 |
| 2005 | 42.27 | 43.98 | 6.91 | 4.49 | 2.30 | 0.06 |
| 2006 | 41.82 | 43.70 | 6.86 | 5.28 | 2.27 | 0.06 |
| 2007 | 43.08 | 43.02 | 6.76 | 4.74 | 2.34 | 0.06 |
| 2008 | 44.55 | 42.57 | 6.69 | 3.71 | 2.42 | 0.06 |
| 2009 | 44.88 | 42.48 | 6.67 | 3.47 | 2.44 | 0.06 |
| 2010 | 44.61 | 42.77 | 6.72 | 3.42 | 2.43 | 0.06 |
| 2011 | 45.19 | 42.23 | 6.63 | 3.43 | 2.46 | 0.06 |
| 2012 | 45.70 | 41.95 | 6.59 | 3.23 | 2.49 | 0.06 |
| 2013 | 46.25 | 41.54 | 6.52 | 3.11 | 2.52 | 0.06 |
| 2014 | 46.56 | 41.24 | 6.48 | 3.14 | 2.53 | 0.06 |
| 2015 | 46.59 | 41.25 | 6.48 | 3.09 | 2.53 | 0.06 |
| 2016 | 47.05 | 40.76 | 6.40 | 3.17 | 2.56 | 0.06 |
| 2017 | 47.40 | 40.62 | 6.38 | 2.96 | 2.58 | 0.06 |
| 2018 | 47.74 | 40.33 | 6.33 | 2.95 | 2.60 | 0.06 |
| 2019 | 47.30 | 40.60 | 6.38 | 3.09 | 2.57 | 0.06 |
| 2020 | 47.36 | 41.93 | 6.58 | 3.44 | 0.64 | 0.06 |
| Mean | 44.50 | 42.19 | 6.63 | 4.30 | 2.32 | 0.06 |

This result coincides with the results of existing studies. In existing studies, the feed crops cultivation stage is the largest source of GHG emissions for the swine industry [45], while the manure management stage is the second largest source of GHG emissions [46].

From 2001 to 2020, the proportion of carbon emissions generated from energy consumption during the pig farming stage experienced the most noticeable decline. This proportion has dropped from 9.25% in 2001 to 3.44% in 2020, representing a decrease of 64.86%. The proportions of carbon emissions from the intestinal fermentation stage and manure management stage exhibit a slow decline, with average annual growth rates of −0.36%. Conversely, the proportions of carbon emissions from the feed crop cultivation stage and feed crop transportation and processing stage exhibit a slow increase, with average annual growth rates of 1.08%. The proportion of carbon emissions from the slaughter and processing stage remains relatively stable, staying within the range of 0.05% to 0.06%.

4.3.2. Analysis of the Factors Contributing to Carbon Emissions from the Feed Crop Cultivation Stage

The high proportion of carbon emissions from the feed crop cultivation stage is attributed to two main factors. Firstly, pigs exhibit substantial feed consumption, making them the livestock category with the highest feed consumption in China. According to "China Agricultural Statistical Data" in 2020, pig feed accounted for 43.49% of the total feed production.

Secondly, cultivating feed crops involves various activities such as fertilizer production and transportation, energy consumption in field cultivation, and GHG emissions. In this paper, the influencing factors of carbon emissions in the feed crop cultivation stage include $CO_2$ emissions from agricultural machinery, irrigation energy consumption, agricultural film production, pesticide production, and $CO_2$ emissions from urea application. Notably, the largest contributor to emissions in this stage is synthetic nitrogen fertilizers.

Research indicates that synthetic nitrogen fertilizers account for 44% to 79% of the total life cycle carbon emissions for crops such as rice, wheat, and maize [47]. Agricultural inputs have long been a significant source of energy consumption and environmental pollution in

China [45,48], with the excessive use of nitrogen fertilizers being a major contributor. The excessive consumption of nitrogen fertilizers surpasses the actual requirements for crop growth, leading to adverse environmental impacts during the cultivation of feed crops [49].

4.3.3. Analysis of the Factors Contributing to Carbon Emissions from the Manure Management Stage

Two main reasons contribute to the high proportion of carbon emissions from the manure management stage. Firstly, pig manure has higher organic matter, and compared to the manure of ruminant animals, pig manure emits more greenhouse gases during the fermentation process [45,46]. Studies by Xu Xingying et al. found that carbon emissions from pig manure accounted for 84.09% of annual methane emissions from manure management, far exceeding emissions from ruminant animal manure [50]. Secondly, there is a high proportion of small-scale breeders in China's pig farming households. These small-scale breeders invest most of their funds in the construction of pigsties, while the methods for handling pig manure are comparatively simple and outdated. Many small-scale farms adopt methods like natural composting and thick bedding, utilizing pig manure as fertilizer in the fields. This directly increases carbon emissions from the manure management stage.

## 5. Decomposition and Analysis of Factors Influencing Carbon Emissions in the Swine Industry

### 5.1. Decomposition of Factors Influencing National Carbon Emissions in the Swine Industry

Based on the LMDI additive decomposition model, this paper calculates the cumulative contribution of production efficiency, industry structure, economic development, urbanization level, and population size to the incremental carbon emissions in the national swine industry from 2001 to 2020. This analysis aimed to assess the impact of each factor on the carbon emissions in the swine industry (Table 8).

**Table 8.** Decomposition of Factors Influencing National Carbon Emissions in the Swine Industry (in $10^4$ tonnes $CO_2$-eq).

| Year | Production Efficiency $\Delta EI$ | Industry Structure $\Delta AI$ | Economic Development $\Delta CI$ | Urbanization Level $\Delta UR$ | Population Size $\Delta P_T$ | Total Effect $\Delta C$ |
|---|---|---|---|---|---|---|
| 2002 | −383.01 | −176.22 | 406.19 | −2.81 | 50.10 | −105.74 |
| 2003 | −457.72 | −480.09 | 1623.38 | −378.81 | 99.93 | 406.70 |
| 2004 | −5904.88 | 319.10 | 6692.99 | −476.25 | 150.38 | 781.34 |
| 2005 | −8514.58 | 932.97 | 8714.62 | −442.76 | 199.99 | 890.23 |
| 2006 | −8493.89 | 591.97 | 9533.82 | −581.02 | 249.34 | 1300.22 |
| 2007 | −7752.19 | −22.93 | 7867.13 | −670.05 | 265.63 | −312.42 |
| 2008 | −11,056.72 | 1009.88 | 10,801.46 | −803.37 | 316.09 | 267.34 |
| 2009 | −12,803.63 | 1210.74 | 12,841.79 | −958.33 | 363.26 | 653.83 |
| 2010 | −10,790.44 | −411.37 | 13,167.37 | −1601.31 | 405.05 | 769.29 |
| 2011 | −11,361.57 | −173.73 | 13,691.95 | −1728.90 | 447.09 | 874.84 |
| 2012 | −13,960.79 | 487.56 | 16,473.02 | −2020.77 | 505.16 | 1484.18 |
| 2013 | −13,810.85 | 211.01 | 17,197.30 | −2280.22 | 558.97 | 1876.21 |
| 2014 | −14,020.16 | 6.90 | 18,216.73 | −2532.86 | 617.59 | 2288.19 |
| 2015 | −13,263.67 | −825.56 | 17,927.53 | −2721.93 | 643.61 | 1759.97 |
| 2016 | −13,631.51 | −676.06 | 18,266.60 | −2961.07 | 691.85 | 1689.82 |
| 2017 | −14,741.00 | 165.28 | 19,178.30 | −3269.78 | 753.55 | 2086.36 |
| 2018 | −13,843.64 | −357.02 | 19,087.40 | −3512.58 | 790.34 | 2164.50 |
| 2019 | −11,896.61 | −2090.49 | 16,401.08 | −3271.10 | 717.34 | −139.79 |
| 2020 | −14,396.41 | −1005.02 | 18,579.69 | −3968.29 | 777.36 | −12.67 |
| Cumulative Contribution | −201,083.26 | −1283.07 | 246,668.34 | −34,182.22 | 8602.63 | 18,722.41 |

### 5.1.1. Factors Inhibiting Carbon Emissions in the Swine Industry

From the cumulative contribution of each influencing factor, three factors—production efficiency, urbanization level, and industry structure—have inhibited carbon emissions in China's swine industry. Compared to the base year of 2001, these three factors have reduced carbon emissions by around 2365.49 million tonnes from 2002 to 2020, about −12.63 times the total effect ($\Delta C$) of national carbon emission increment.

(1) Production Efficiency Factor: The cumulative contribution of the production efficiency factor is significantly greater than those of the industry structure and urbanization

level, emphasizing its crucial role in reducing carbon emissions in the swine industry. In the existing studies, Yao Chengsheng et al. found that production efficiency is the most important factor inhibiting carbon emissions from China's livestock industry [20].

From 2001 to 2020, the production efficiency factor, measured by carbon emissions per unit output value in pig farming, has significantly decreased from 371 g/CNY to 59 g/CNY. Compared to the base year, the improvement in production efficiency has led to a cumulative reduction of 2010.83 million tonnes of carbon emissions, about −10.74 times the national carbon emission increment. Assuming other factors remain constant, the improvement in production efficiency will result in a reduction in carbon emissions from the swine industry of about 105.83 million tonnes per year. This emphasizes the importance of production efficiency in reducing carbon emissions.

(2) Urbanization Level Factor: The cumulative contribution of the urbanization level factor is also negative. Though its numerical magnitude is only 17% of the production efficiency factor, it still exerts a significant inhibitory effect on carbon emissions. From 2001 to 2020, China's urbanization level has steadily increased, with the rural population decreasing from 60.09% in 2001 to 36.16% in 2020. As the percentage of the rural population diminishes, the labor force engaged in the primary industry, particularly animal husbandry, also decreases. The rise in urbanization level has resulted in a cumulative reduction of 341.82 million tonnes of carbon emissions in the swine industry, accounting for −182.57% of the national carbon emission increment. Assuming other factors remain constant, the improvement in urbanization level will reduce carbon emissions from the swine industry by 17.99 million tonnes per year. This underscores the significant role of urbanization level factor in carbon emission reduction in the swine industry.

(3) Industry Structure Factor: Although the negative cumulative contribution of the industry structure factor is the smallest, it has reduced carbon emissions by 62.18 million tonnes over the study period, constituting −33.21% of the swine industry's carbon emission increment ($\Delta C$). This underscores the non-negligible role of optimizing the industry structure in restraining carbon emissions in the swine industry. Furthermore, from 2001 to 2020, the highest value of the industry structure factor (i.e., the proportion of the pig farming output value in the livestock industry output value) was 52% in 2009, and the lowest was 34% in 2019, with an average of 45%. As the industry structure factor increases, its dampening effect on carbon emissions from the swine industry weakens. When the critical threshold is surpassed, the industry structure factor may even increase carbon emissions. This result coincides with the existing literature studies [20], and the critical threshold is estimated to fall between 44.69% and 44.86%.

5.1.2. Factors Promoting Carbon Emissions in the Swine Industry

From 2001 to 2020, the economic development factor and population size factor have had a positive cumulative contribution, which has promoted carbon emissions in China's swine industry. During the study period, these two factors contributed to an increase of 2552.71 million tonnes in carbon emissions, about 13.63 times the carbon emission increment ($\Delta C$) in the swine industry. Furthermore, the cumulative contribution of economic development factors is 28.67 times that of population size factors.

(1) Economic Development Factor: The economic development factor (ratio of total livestock production value to total rural population) is growing at a significant rate, from CNY 0.61 thousand per person in 2001 to CNY 6.62 thousand per person in 2020—a nearly 11-fold increase in just 20 years. The rapid improvement in economic development has increased 2466.68 million tonnes of carbon emissions from China's swine industry. Assuming other factors remain constant, the elevation in economic development will result in an annual increase of 129.83 million tonnes in carbon emissions from China's swine industry. In the existing studies, Liu Yang and Liu Hongbin used the LMDI model to analyze the influencing factors of agricultural carbon emissions in Shandong Province, and found that economic development is the main factor for the increase in agricultural carbon emissions [39]. As China is a major producer and consumer of pork, with the

swine industry's scale and consumption continued rising alongside domestic economic development, the economic development factor is likely to maintain a dominant position in increasing carbon emissions in the swine industry.

(2) Population Size Factor: Compared to the economic development factor, the promoting effect of the population size factor on carbon emissions in the swine industry may seem negligible, but this is not the case. From 2001 to 2020, China's population has grown consistently, increasing from 1.28 billion to 1.41 billion. Population growth inevitably drives the domestic demand for pork, subsequently increasing carbon emissions from the swine industry. The cumulative contribution of the population size factor to carbon emissions in the swine industry from 2001 to 2020 is 86.03 million tonnes. Assuming other factors remain constant, this equals an annual increase of 4.53 million tonnes of carbon emissions. Cao Lihong et al. similarly concluded that the number of laborers employed in the pig farming industry has a promoting effect on the carbon emissions of the pig farming industry, assuming other factors remain unchanged [21].

However, despite China's large population base, the low population growth rate of 0.53% during 2001–2020, influenced by the family planning policy, is a reason for the low promoting effect of the population size factor on carbon emissions in the swine industry. Although the two-child policy has now been implemented, the birth rate in China has no significant improvement. As of 2021, China officially entered a phase of negative population growth. If population reduction continues in the future, the impact of the population size factor on carbon emissions in the swine industry may shift from positive to negative.

*5.2. Decomposition of Carbon Emission Influencing Factors in Provincial Swine Industries*

This paper decomposes the influencing factors of carbon emissions in the swine industry across 30 provinces and municipalities in China, and calculates the cumulative contribution of five factors on the carbon emissions in the swine industry across different regions (Table 9).

**Table 9.** Decomposition of Carbon Emission Influencing Factors in Provincial Swine Industry (2001–2020, in $10^4$ tonnes $CO_2$-eq).

| Region | Production Efficiency $\Delta EI$ | Industry Structure $\Delta AI$ | Economic Development $\Delta CI$ | Urbanization Level $\Delta UR$ | Population Size $\Delta P_T$ |
|---|---|---|---|---|---|
| Beijing | −1208.28 | 96.53 | 620.31 | −200.79 | 269.20 |
| Tianjin | −902.35 | 261.50 | 1114.85 | −563.73 | 206.05 |
| Hebei | −12,313.69 | −165.41 | 14,530.43 | −2407.59 | 609.89 |
| Shanxi | −1764.50 | −88.62 | 2646.35 | −543.80 | 123.71 |
| Inner Mongolia | −2488.72 | −1888.11 | 4887.74 | −610.48 | 83.68 |
| Liaoning | −4165.50 | −695.51 | 8192.11 | −943.46 | 147.33 |
| Jilin | −4487.85 | 21.67 | 4917.88 | 4.41 | 24.07 |
| Heilongjiang | −4324.10 | −231.10 | 6304.43 | −201.03 | −33.05 |
| Shanghai | −1125.20 | 444.09 | −95.94 | −58.48 | 240.28 |
| Jiangsu | −8828.47 | 610.20 | 9919.32 | −2597.11 | 443.42 |
| Zhejiang | −5694.67 | 1131.14 | 4614.06 | −845.31 | 566.34 |
| Anhui | −10,185.19 | 1165.95 | 10,798.24 | −1357.35 | −173.56 |
| Fujian | −4716.42 | 84.61 | 5604.61 | −555.49 | 329.06 |
| Jiangxi | −7469.97 | 241.95 | 9755.85 | −1540.74 | 403.85 |
| Shandong | −13,463.22 | 2999.11 | 13,418.85 | −1961.78 | 628.62 |
| Henan | −13,162.98 | −250.56 | 19,490.46 | −2912.97 | −51.99 |
| Hubei | −9143.43 | 1016.03 | 12,471.78 | −1079.00 | −197.70 |
| Hunan | −20,440.44 | −894.38 | 24,441.16 | −3585.3 | 73.32 |
| Guangdong | −11,784.67 | 761.28 | 13,896.15 | −4406.90 | 2293.65 |
| Guangxi | −8774.44 | −1714.65 | 12,540.07 | −1480.00 | 15.45 |
| Hainan | −1099.00 | 111.27 | 1592.64 | −201.18 | 93.03 |
| Chongqing | −6281.38 | −866.23 | 8853.56 | −1886.27 | −199.56 |
| Sichuan | −24,483.77 | −1866.05 | 28,535.45 | −1520.62 | −770.87 |
| Guizhou | −6347.96 | −662.01 | 7631.15 | −411.50 | −190.49 |
| Yunnan | −8410.88 | −967.01 | 12,548.13 | −1471.18 | 517.07 |
| Shaanxi | −3144.37 | 40.07 | 4444.77 | −679.13 | 69.21 |
| Gansu | −1742.08 | −431.62 | 2495.07 | −229.54 | 8.92 |
| Qinghai | −354.06 | −97.28 | 530.44 | −58.66 | 23.59 |
| Ningxia | −255.58 | −168.89 | 452.66 | −60.88 | 36.36 |
| Xinjiang | −801.63 | 98.39 | 1176.75 | −49.36 | 132.27 |

5.2.1. Production Efficiency as a Suppressive Factor of Carbon Emissions in Provincial Swine Industries

The production efficiency factor exhibited a restraining effect on carbon emissions in the swine industry across all 30 provinces and municipalities from 2001 to 2020. The cumulative contribution of this factor to carbon emission reduction is negative for all regions, making it the most significant factor suppressing carbon emissions. In descending order of impact, the top five regions with the most significant carbon emission reductions due to production efficiency are Sichuan, Hunan, Shandong, Henan, and Hebei. These five regions have accumulated carbon reduction contributions exceeding 100 million tonnes, with Sichuan and Hunan surpassing even 200 million tonnes, showcasing remarkable effectiveness. Additionally, these five provinces are major grain-producing areas in China and have high pig farming volumes, consequently ranking among the top in the swine industry's carbon emissions. As the swine industry shifts toward large-scale and intensive operations, these regions have concentrated their advantageous resources, accumulated substantial breeding experience, enhanced breeding technologies, and established more large-scale pig farming waste treatment facilities. These efforts have significantly reduced carbon emissions from the local swine industry.

While, in ascending order of impact, the last four regions are Tianjin, Xinjiang, Qinghai, and Ningxia. Tianjin is an economically developed grain distribution area, but Xinjiang, Qinghai, and Ningxia are economically less developed western regions. For the former, agriculture is not the primary economic contributor, and local governments often prioritize the development of secondary and tertiary industries. Consequently, pig farming scales are relatively small, resulting in lower swine industry carbon emissions, and the impact of production efficiency on carbon emissions is limited. For the latter, due to lower economic development levels, smaller swine industry scales, and relatively outdated production technologies, traditional free-range farming practices persist, leading to inefficient treatment of pig farming waste. Therefore, the production efficiency factor has a weak dampening effect on carbon emissions from the swine industry in these regions.

5.2.2. Regional Disparities in the Impact of Industry Structure on Carbon Emissions in Provincial Swine Industries

The impact of the industry structure factor on carbon emissions in the swine industry varies significantly across regions. In terms of quantity, 15 regions exhibit a negative cumulative contribution of the industry structure factor and 16 regions show a positive cumulative contribution. However, in terms of effectiveness, eight regions with a negative cumulative contribution have achieved cumulative carbon reductions exceeding 6.5 million tonnes. Inner Mongolia, Sichuan, and Guangxi have surpassed 15 million tonnes in cumulative carbon reduction. Among the regions with a positive cumulative contribution, only four regions have cumulative carbon emissions exceeding 6.5 million tonnes, with Shandong surpassing 15 million tonnes. Therefore, the impact of the industry structure factor on carbon emissions in provincial swine industries has an inhibiting effect.

In regions where the industry structure factor suppresses local swine industry carbon emissions, the values of this factor have notably declined in Inner Mongolia, Sichuan, Guangxi, Yunnan, Hunan, Chongqing, Liaoning, Guizhou, Gansu, Ningxia, and Qinghai. Alongside adjustments in industry structure, the proportion of local pig farming output in the livestock industry's output has decreased, leading to a substantial reduction in carbon emissions. Conversely, in regions where the industry structure factor promotes carbon emissions in the swine industry, the values of this factor have shown a fluctuating upward trend in Shandong, Anhui, Zhejiang, Hubei, Jiangsu, Shanghai, Tianjin, Xinjiang, Beijing, and Jilin. Under this trend, adjustments in industry structure have led to an increase in local swine industry carbon emissions.

### 5.2.3. Economic Development as a Primary Driver of Carbon Emissions in Provincial Swine Industries

Economic development is the predominant factor driving carbon emissions in the swine industry across the 29 provinces and municipalities. Among these, Sichuan demonstrates the largest cumulative contribution at 285.35 million tonnes, while Ningxia exhibits the smallest at only 4.53 million tonnes. In descending order of impact, the top ten regions are Sichuan, Hunan, Henan, Hebei, Guangdong, Shandong, Yunnan, Guangxi, Hubei, and Anhui. Compared to 2001, these 10 regions have all experienced carbon emission increases exceeding 100 million tonnes due to the rise in economic levels, with Sichuan and Hunan surpassing 200 million tonnes in carbon emission increments. This list aligns with the top ten national rankings in pig production. Except for Yunnan, Guangdong, and Guangxi, the other seven regions are major grain-producing areas in China. Given the higher profitability of livestock farming compared to cultivation, farmers in these regions tend to expand pig farming to enhance economic returns, resulting in an increase in local swine industry carbon emissions.

For economically developed regions like Shanghai, where the pig farming scale is typically smaller, the improvement in economic returns for local farmers is not solely reliant on increasing livestock and poultry farming. Farmers can diversify into non-agricultural sectors for additional income, contributing to the limited impact of economic development on carbon emissions in the local swine industry.

### 5.2.4. Urbanization Level as a Suppressive Factor of Carbon Emissions in Provincial Swine Industries

The urbanization level is a suppressive factor in carbon emissions across 29 provinces and municipalities in the swine industry. Among these, Guangdong exhibits the most effective suppression, with a carbon reduction of 44.07 million tonnes, while Xinjiang shows the least effective suppression at only $49 \times 10^4$ tonnes. In descending order of impact, the top ten regions are Guangdong, Hunan, Henan, Jiangsu, Hebei, Shandong, Chongqing, Jiangxi, Sichuan, and Guangxi. Compared to 2001, these 10 regions have reduced carbon emissions from the swine industry exceeding 10 million tonnes due to the improvement in urbanization levels, with Guangdong, Hunan, Henan, Jiangsu, and Hebei surpassing 20 million tonnes in carbon reduction. Among the top ten provinces, seven are major grain-producing areas, namely, Hunan, Henan, Jiangsu, Hebei, Shandong, Jiangxi, and Sichuan. The remaining three, while not classified as major grain-producing areas, boast significant pig farming scales, maintaining annual slaughter volumes of around 20 million heads.

In the ten regions mentioned above, local agriculture and animal husbandry are relatively well developed. Regions with well-established agriculture and livestock farming witness relatively lower levels of urbanization. As the urbanization level rises, people migrate from rural to urban areas, resulting in a decline in the rural populations and the proportion of the labor force engaged in primary industries. From 2001 to 2020, 30 provinces and municipalities reduced the proportion of rural populations, particularly in the top ten regions with optimal carbon emission suppression. The average decline in the proportion of rural populations in these regions was 27%, surpassing the national average reduction of 21.36%. This suggests that the improvement in urbanization level has facilitated the significant migration of agricultural labor to non-agricultural sectors, effectively restraining carbon emissions in the swine industry.

### 5.2.5. The Influence of Population Size Factors on Carbon Emissions in Provincial Swine Industries Is Predominantly Promotional

Among the 30 provinces and municipalities, 23 regions experience a promotional effect on local swine industry carbon emissions due to population size factors. Of these, 11 regions have a cumulative contribution exceeding 2 million tonnes, and 7 regions surpass 4 million tonnes. In descending order of impact, the top ten regions are Guangdong, Shandong, Hebei, Zhejiang, Yunnan, Jiangsu, Jiangxi, Fujian, Beijing, and Shanghai.

Among these regions, Guangdong, Zhejiang, Fujian, Beijing, and Shanghai are major grain-consuming areas, while Shandong, Hebei, Jiangsu, and Jiangxi are major grain-producing areas. As the standard of living continues to rise, the demand for meat products, especially pork, steadily increases, and the continuous expansion of the population further widens the domestic pig market demand. For major grain-consuming areas, the developed local economy attracts an influx of population from other regions, increasing carbon emissions in the local swine industry. In major grain-producing areas with well-established agricultural economies, the population growth depends on agricultural development, which contributes to carbon emissions in the local swine industry.

*5.3. Decomposition of Carbon Emission Influencing Factors in Production Stages of the Swine Industry*

Based on the LMDI additive decomposition model, this paper conducts a factor decomposition of carbon emissions in various production stages of the swine industry (Table 10).

**Table 10.** Decomposition of Carbon Emission Influencing Factors in Production Stages of the Swine Industry (2001–2020, in $10^4$ tonnes $CO_2$-eq).

| Production Stage | Production Efficiency $\Delta EI$ | Industry Structure $\Delta AI$ | Economic Development $\Delta CI$ | Urbanization Level $\Delta UR$ | Population Size $\Delta P_T$ | Total Effect $\Delta C$ |
|---|---|---|---|---|---|---|
| Feed Crop Cultivation | −76,955.04 | −607.97 | 105,137.70 | −14,660.73 | 3675.89 | 16,589.86 |
| Feed Crop Transport and Processing | −4184.96 | −33.06 | 5717.58 | −797.28 | 199.90 | 902.19 |
| Intestinal Fermentation | −13,387.17 | −77.01 | 16,270.59 | −2244.80 | 566.36 | 1127.97 |
| Manure Management | −85,250.76 | −490.39 | 103,612.66 | −14,295.09 | 3606.63 | 7183.05 |
| Energy Consumption in Pig Farming | −19,951.67 | −57.47 | 14,369.21 | −1953.69 | 498.04 | −7095.59 |
| Slaughter and Processing | −107.81 | −0.67 | 137.63 | −19.01 | 4.79 | 14.93 |

Firstly, economic development is the predominant factor in increasing carbon emissions in the six stages of the swine industry. Conversely, production efficiency is the major factor restraining the increase in carbon emissions in these six stages.

Secondly, the cumulative contribution for the incremental carbon emissions in each production stage under the influence of five factors consistently exhibits the same change direction. Production efficiency, industry structure, and urbanization level inhibit carbon emissions in the six production stages. In contrast, economic development and population size promote carbon emissions in each production stage.

Thirdly, compared to 2001, there are variations in the carbon emission increases (total effect $\Delta C$) across different stages of the swine industry from 2002 to 2020. Except for the energy consumption in the pig farming stage, carbon emission increases in the remaining five stages are positive, aligning with the overall fluctuation in national swine industry carbon emissions. The reduction in carbon emissions from energy consumption in pig farming can be attributed to a shift in heating methods within pig farms. Temperature influences pig growth, and to ensure optimal growth rates, larger-scale pig farms often use coal, electricity, or natural gas for heating. However, with increased efforts to combat air pollution, governments have introduced environmental protection policies to solve the problem of coal-based heating. According to data from the "China Animal Husbandry and Veterinary Yearbook", the average coal consumption per pig has decreased from 12.32 kg in 2001 to 1.10 kg in 2020. Meanwhile, the average electricity consumption per pig has increased from 4.87 kWh in 2001 to 9.04 kWh in 2020. Consequently, the shift in heating methods has reduced carbon emissions in the energy consumption in pig farming stage.

## 6. Conclusions and Policy Recommendations

### *6.1. Key Findings*

From the perspective of the industry chain, this paper utilized the LCA method to assess and analyze the variations and characteristics of carbon emissions in China's swine industry from 2001 to 2020 at the national, provincial, and production stage levels. Based on the Kaya model, this paper constructed the LMDI additive decomposition model to analyze the influencing factors of carbon emissions in the swine industry from the national, provincial, and production stage levels. The key findings are as follows:

(1) The carbon emissions in China's swine industry exhibited a general trend of "slow growth—sharp decline—fluctuating rise—fluctuating decline". The growth in demand for pork consumption, which drives an increase in pig supply, is the root cause of the rise in carbon emissions in the swine industry.

From 2001 to 2020, national swine industry carbon emissions peaked at 101.08 million tonnes in 2014 and reached a low of 75.07 million tonnes in 2007. Also, it experienced significant setbacks in 2007 and 2019, with a decline of up to 17.68% and 26.12%, respectively.

Carbon emissions in China's swine industry during the period from 2001 to 2006 witnessed slow growth, attributed to the structural adjustments in the swine industry and the influence of agricultural policy directions. The stage from 2006 to 2007 marked a sharp decline, mainly due to the lack of confidence in the market among breeders caused by swine epidemics. The period from 2007 to 2018 saw a fluctuating rise, driven by an increase in the proportion of large-scale swine farming and innovations in swine farming practices. Finally, the stage from 2018 to 2020 experienced a fluctuating decline, influenced by the outbreak of African swine fever, the implementation of the "Environmental Protection Tax Law", and the impact of the COVID-19 pandemic on the swine market.

(2) There are significant regional differences in carbon emissions from the provincial swine industries. Taking the average carbon emissions (2.60 million tonnes) and average carbon intensity (0.3409 tonnes/head) of the swine industry in 2020 as a benchmark, regions with high carbon emissions and carbon intensities like Jiangsu, Anhui, and Henan are overly pursuing the economic benefits of the swine industry and neglecting carbon emissions, while regions with high carbon emissions and low carbon intensities can be the focus of the low-carbon development of the swine industry.

Among the 30 provinces and municipalities, the highest carbon intensity is in Xinjiang (0.38 tonnes/head), and the lowest is in Yunnan (0.332 tonnes/head). Taking the average values of carbon emissions and carbon intensity of the swine industry as the division standard, the provinces and municipalities can be divided into four types: double high carbon emissions and carbon intensity, high carbon emissions and low carbon intensity, low carbon emissions and high carbon intensity, and double low carbon emissions and carbon intensity, of which the most noteworthy are the first two types.

Firstly, Jiangsu, Anhui, and Henan have double high carbon emissions and carbon emissions intensity, which indicates that they over-pursued the economic benefits of the swine industry and neglected the environmental benefits. Secondly, Hebei, Liaoning, Inner Mongolia, Jiangxi, Shandong, Hubei, Hunan, Guangdong, Guangxi, Sichuan, and Yunnan have high carbon emissions and low carbon intensity, which indicates that they perform better in carbon emission reduction in the swine industry, and can be used as the key planning provinces for low-carbon development of the swine industry.

(3) The cultivation of feed crops and manure management were identified as the primary sources of carbon emissions in the swine industry.

The average proportion of carbon emissions from each production stage in the total carbon emissions of the swine industry from 2001 to 2020 is as follows: feed crop cultivation (44.50%), manure management (42.19%), intestinal fermentation (6.63%), energy consumption in pig farming (4.30%), feed crop transport and processing (2.32%), and slaughter and processing (0.06%).

The high proportion of carbon emissions in the feed crop cultivation stage is attributed to two main reasons: first, the substantial consumption of pig feed; second, the energy

consumption in the stages of feed crop cultivation, fertilizer production and transportation, and agricultural field operations, which results in substantial carbon emissions. Meanwhile, the high carbon emissions in the manure management stage can be explained by two factors: first, the significantly higher carbon emissions generated by the fermentation of pig manure compared to ruminant animals; second, the relatively outdated manure management practices adopted by many pig farms.

(4) Production efficiency, urbanization level, and industry structure have inhibitory effects on carbon emissions in China's swine industry, while economic development and population size exert promoting effects.

Irrespective of the level—national, provincial, or production stage—economic development stands out as the primary factor promoting carbon emissions, while production efficiency is the foremost factor inhibiting carbon emissions. This underscores the importance of improving production efficiency to curb carbon emissions in the swine industry.

*6.2. Policy Recommendations*

To advance the low-carbon development of the swine industry, this paper proposes the following policy recommendations to mitigate carbon emissions in China's swine industry:

(1) The government can enhance the production efficiency of the swine industry and restrain carbon emissions by strengthening environmental regulations, formulating a long-term carbon reduction plan for the swine industry, and increasing investment in funds and talent for technological research and development.

Firstly, in the process of swine industry development, maximizing economic benefits often leads to the expansion of breeding scale, neglecting the necessity of environmental pollution control. The government should implement robust environmental control measures, formulate and implement a comprehensive long-term plan for carbon reduction in the swine industry, regulate domestic pig supply based on pig breeding demands and domestic land carrying capacity, and strictly control the number of domestic pig breeding.

Secondly, set strict targets for carbon emissions and carbon intensity, improve the carbon reduction monitoring mechanism in the swine industry, and ensure effective implementation of environmental control measures.

Lastly, the government can enhance production efficiency in the swine industry by increasing the proportion of large-scale breeding, investing in low-carbon pig farming technologies, improving the technical skills of breeding households, and attracting high-quality talents, effectively reducing the unit carbon emissions of the swine industry.

(2) The central government can optimize the layout of the swine industry by designating key areas for carbon reduction and establishing carbon reduction demonstration bases, provincial governments can formulate specific control measures based on local conditions.

Firstly, the central government should clearly define key areas for carbon reduction in the domestic swine industry, such as Jiangsu, Anhui, and Henan. Initiate efforts to build a carbon emissions assessment system for the swine industry and establish carbon reduction demonstration bases, contributing to a nationwide plan for carbon reduction in the swine industry and optimizing its industrial layout.

Secondly, differentiating carbon reduction policies based on local conditions is crucial for suppressing local swine industry carbon emissions. Building upon the national plan, provincial governments should establish more specific and scientifically reasonable carbon reduction targets based on local swine industry conditions and resource endowments. The aim is to enhance low-carbon farming technologies, implement relevant responsibilities, and clear incentives and penalties to ensure rapid and sustainable development of the local swine industry while effectively addressing carbon emissions.

(3) Carbon reduction in the swine industry should focus on feed crop cultivation and manure management, which can be achieved by implementing measures such as breeding superior varieties, enhancing techniques in fertilization and manure treatment, improving agricultural machinery, and increasing resource utilization efficiency.

In the feed crop cultivation stage, the government can promote the use of environmentally friendly materials such as degradable films and packaging bags, and introduce technologies such as AI robots, big data, and intelligent manufacturing to enhance the mechanical equipment level and resource utilization efficiency in feed crop cultivation. Promoting the cultivation of superior feed varieties, increasing straw return, and adopting advanced fertilization processes to improve fertilizer utilization can effectively reduce carbon emissions in the feed crop cultivation stage.

In the manure management stage, the government can encourage and guide breeding households to adopt low-carbon manure treatment technologies, such as the fertilization and resource utilization of pig manure, through policy subsidies and assistance, so as to build a sustainable development pattern of the swine industry.

**Author Contributions:** Conceptualization, Y.Y. and Q.L.; methodology, Y.B.; validation, Q.L.; formalanalysis, Q.L.; resources, Y.C.; datacuration, T.N.; writing-original draft preparation, Q.L.; writingreview and editing, Y.Y., Q.L. and E.F.; visualization, Y.C. and T.N.; funding acquisition, Y.Y.; Supervision, Y.Y. All authors have read and agreed to the published version of the manuscript.

**Funding:** This research was fund by Ningbo Institute of Urban Civilization "Investigation on the Environmental and Sanitation Conditions of Rural Villages in Ningbo City and Research on the Improvement and Construction (CSWM202211)"; Advanced Humanities and Social Sciences Cultivation Project in Ningbo University in 2022 (Pro-phase Project of Cultivation) "Research on Synergistic Effect of Reducing Pollution and Carbon (XPYQ22001)"; Zhejiang Philosophy and Social Science Planning Project "Research on construction mechanism and path of Ecological Civilization High-land in Zhejiang"; General Scientific Research Project of Zhejiang Education Department (Postgraduate Special Project) "Research on the measurement and influencing factors of carbon emissions in the swine industry from the perspective of the industry chain (Y202250003)".

**Institutional Review Board Statement:** Not applicable.

**Informed Consent Statement:** Not applicable.

**Data Availability Statement:** Data are contained within the article.

**Conflicts of Interest:** The authors declare no conflict of interest.

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
