# Peer review of "Research on the Measurement and Influencing Factors of Carbon Emissions in the Swine Industry from the Perspective of the Industry Chain"

_sustainability, doi:10.3390/su16052199_

Round 1
Reviewer 1 Report
Comments and Suggestions for Authors
The authors proposed suggestions are including strengthening environmental control, formulating long-term plans for carbon emission reduction, delineating key areas and demonstration bases for carbon emission reduction, enhancing expertise in fertilizer application and manure treatment, and improving the level of agricultural machinery and equipment. All the suggestion are valuable, but there are also some issues that exist.
1、Literature Review is too long, Please abbreviate section 1.2.
2、The paper lacks a discussion section. What are the differences between the research of other scholars and this study?
3. Where is the author's original data source? What are the characteristics of the raw data? For example, maximum, minimum, and average values.
Comments on the Quality of English LanguageI consider that the English language should be moderately revised.
Author Response
Response to Reviewers
The authors proposed suggestions are including strengthening environmental control, formulating long-term plans for carbon emission reduction, delineating key areas and demonstration bases for carbon emission reduction, enhancing expertise in fertilizer application and manure treatment, and improving the level of agricultural machinery and equipment. All the suggestion are valuable, but there are also some issues that exist.
Point 1: Literature Review is too long, Please abbreviate section 1.2.
Response 1: Thank you for your suggestions. We have reduced the literature review from 5,382 to 4,065 characters, as detailed in lines 110-167 of the revised manuscript.
Point 2: The paper lacks a discussion section. What are the differences between the research of other scholars and this study?
Response 2: Thank you for your suggestions. Regarding the differences between other scholars' studies and this study, which have been discussed in the Literature Review section of this paper, the paragraph structure of the Literature Review has now been further adjusted in order to make the relevant content more prominent, as detailed in the red-lettered paragraphs in lines 150-167 of the revised manuscript.
Point 3: Where is the author's original data source? What are the characteristics of the raw data? For example, maximum, minimum, and average values.
Response 3: Thank you for your suggestions. We are very sorry for our careless mistakes. The source of the original data is on page 10 of this paper, and the bibliographies of cited sources have now been added in the footnotes and references. A brief summary of the original data has also been added to the paper, as detailed in Table 4 on page 11 and Table 5 on page 14 of the revised manuscript.

Reviewer 2 Report
Comments and Suggestions for Authors
The paper deals with an important topic- GHG emission reduction options in the swine industry from the perspective of the industry chain. The paper applies LCA and other advanced methods to conduct this research however, more background and justification should be provided for carbon emission coefficients selected for each stage of the swine industry and provided in Table. Usually, the LCA studies are using harmonized data on GHG emissions like ecoinvent. In this paper GHG emission factors are taken from various sources and are obtained by applying different approaches and methods therefore I am afraid about validity of results of conducted study.
Reviewer 3 Report
Comments and Suggestions for Authors
1. Brief summary
The paper addresses a relevant and significant topic. This manuscript uses the Life Cycle Assessment (LCA) method to estimate carbon emissions in the Chinese swine industry at both national and provincial levels for the period 2001-2020, adopting an industry chain perspective. The study involves a temporal characteristic analysis, spatial disparity analysis, and production stage contribution analysis of the estimated results. Additionally, the paper evaluates the influencing factors of carbon emissions in the Chinese swine industry. The novelty of this paper lies in its industry chain approach, specifically focusing on the pig industry, which includes feed production, pig production, and slaughtering and processing.
2. General concept comments
The manuscript is clear and presented in a well-structured manner. Authors correctly present the problems, the scientific gaps, and research objectives. The cited references are relevant, with a majority within the last decade. It doesn’t include an excessive number of self-citations. The paper scientifically sounds and the research design is appropriate. However, a weakness can be highlight. The authors apply deterministic models with basic data sourced from various databases, lacking statistical analysis. Incorporating correlation and regression analyses to evaluate relationships between the basic data would enhance the scientific robustness of the paper. Despite this weakness the interpretation of results based on the applied deterministic models is appropriate and understandable. The discussion, however, requires improvement, particularly in relation to integrating literature insights alongside the presented results. Conclusions and recommendations are consistent with the evidence presented in the paper.
3. Specific comments
Line 200-201: Authors wrote “some researchers”, but references are missing.
Line 223: “scholars commonly use…” Some references are recommended here.
Line 345-346: “Common methods for decomposing carbon emission factors include the IPAT model, STIRPAT model, and Kaya model.” Some references are recommended here.
Subchapter 3.3.: It is recommended to add the bibliographies of cited sources, databases in the text and in the References in order that basic data can be available by other researchers.
Table 5: It is recommended to add the meaning of each indicator (abbreviation) in a footnote below the table.
Round 2
Reviewer 1 Report
Comments and Suggestions for Authors
The author has made significant improvements in the quality of the revised paper. It is hoped that the author will read the entire text thoroughly and make corrections to any inaccuracies.
Comments on the Quality of English LanguageIt is hoped that the author will read the entire text thoroughly and make corrections to any inaccuracies.
Author Response
Response to Reviewers
Point: The author has made significant improvements in the quality of the revised paper. It is hoped that the author will read the entire text thoroughly and make corrections to any inaccuracies.
Response: Thank you for your suggestions. We have carefully read the whole text and made corrections to inaccuracies, as well as refining and deleting some of the redundant sentences, so as to make the presentation in the text more concise, clear and logical. The relevant revisions can be found in the red-lettered part of the revised manuscript.
Reviewer 2 Report
Comments and Suggestions for Authors
The authors have revised their paper based on my comments. I do not have more comments. The paper can be published in current form.
Author Response
Response to Reviewers
Point: The authors have revised their paper based on my comments. I do not have more comments. The paper can be published in current form.
Response: Thank you for your suggestions. We are deeply appreciative.